# Crystallization of polarons through charge and spin ordering transitions in 1T-TaS$_2$

E. S. Bozin[1] ✉, M. Abeykoon[2], S. Conradson [3], G. Baldinozzi [4], P. Sutar[3] & D. Mihailovic [3] ✉

The interaction of electrons with the lattice in metals can lead to reduction of their kinetic energy to the point where they may form heavy, dressed quasi-particles—polarons. Unfortunately, polaronic lattice distortions are difficult to distinguish from more conventional charge- and spin-ordering phenomena at low temperatures. Here we present a study of local symmetry breaking of the lattice structure on the picosecond timescale in the prototype layered dichalcogenide Mott insulator 1T-TaS$_2$ using X-ray pair-distribution function measurements. We clearly identify symmetry-breaking polaronic lattice distortions at temperatures well above the ordered phases, and record the evolution of broken symmetry states from 915 K to 15 K. The data imply that charge ordering is driven by polaron crystallization into a Wigner crystal-like state, rather than Fermi surface nesting or conventional electron-phonon coupling. At intermediate temperatures the local lattice distortions are found to be consistent with a quantum spin liquid state.

Quasi-two dimensional quantum-ordered materials display very diverse and often exotic spin[1–3] and charge ordering behaviour[4], including exciton condensation[5], chiral spin liquids[6] and superconductivity[7,8]. They can also display hidden phases which can be manipulated by pressure[9,10], light[11–15], or doping[16–18]. An important aspect that determines spin ordering behaviour is the competition between Coulomb repulsion and interlayer interactions[19]. A common feature of these systems is that the strength of the electron phonon interaction (EPI) and electron density control the electron kinetic energy. Increasing the EPI leads to bandwidth narrowing, and lattice dressed electron quasiparticles - as originally introduced by Landau in 1933[20]. A further consequence of electron localization is an enhanced Coulomb interaction that leads to correlations, ubiquitous mesoscopic textures[21,22], charge-density wave (CDW) formation[23] and super-conductivity at high critical temperatures in the fluctuation-dominated intermediate coupling regime of layered cuprate superconductors[24]. Recent model calculations[23] applied to wide range of transition metal dichalcogenide (TMD) materials and comparisons with scanning tunneling microsopy experiments highlighted the importance of the Coulomb interaction that lead to ubiquitous charge ordering at magic

filling fractions[23]. Specific features of the band structure may lead to further effects, such as exciton condensation[5] and CDW ordering aided by Fermi surface nesting[25] on top of the seemingly universal charge ordering[23]. Experimental evidence for polarons comes from many different techniques in a very wide variety of materials[26,27], but their role in forming quantum states, such as Wigner crystals, super-conductivity and quantum spin liquids is not experimentally clear, particularly on the transition from the low to high density limit. Direct observation of symmetry-specific polaronic lattice distortions on the low- to high- density limit are crucial for progress in understanding a wide class of quantum materials.

As a prototypical example, the layered dichalcogenide 1T-TaS$_2$ in equilibrium is of interest as a Mott insulator[28], CDW system[4], polaronic superconductor[9] and quantum spin liquid[2]. Additional recent interest in the material comes from the discovery of hidden light- or charge injection- induced mesoscopic metastable phases[12,29] that are important both fundamentally and for applications[30–32]. Structurally, it is a kinetically trapped 1T polymorph of TaS$_2$, with octahedral stacking (Fig. 1a)[4,33–35]. Its deceptively simple electronic band structure in the undistorted phase has a single Ta metal band crossing the Fermi

[1]Condensed Matter Physics and Materials Science Division, Brookhaven National Laboratory, Upton, NY, USA. [2]Photon Sciences Division, Brookhaven National Laboratory, Upton, NY, USA. [3]Dept. of Complex Matter, Jozef Stefan Institute, Ljubljana, Slovenia. [4]Centralesupélec, CNRS, SPMS, Université Paris-Saclay, bât Eiffel, Gif-sur-Yvette, Île-de-France, France. ✉e-mail: bozin@bnl.gov; dragan.mihailovic@ijs.si

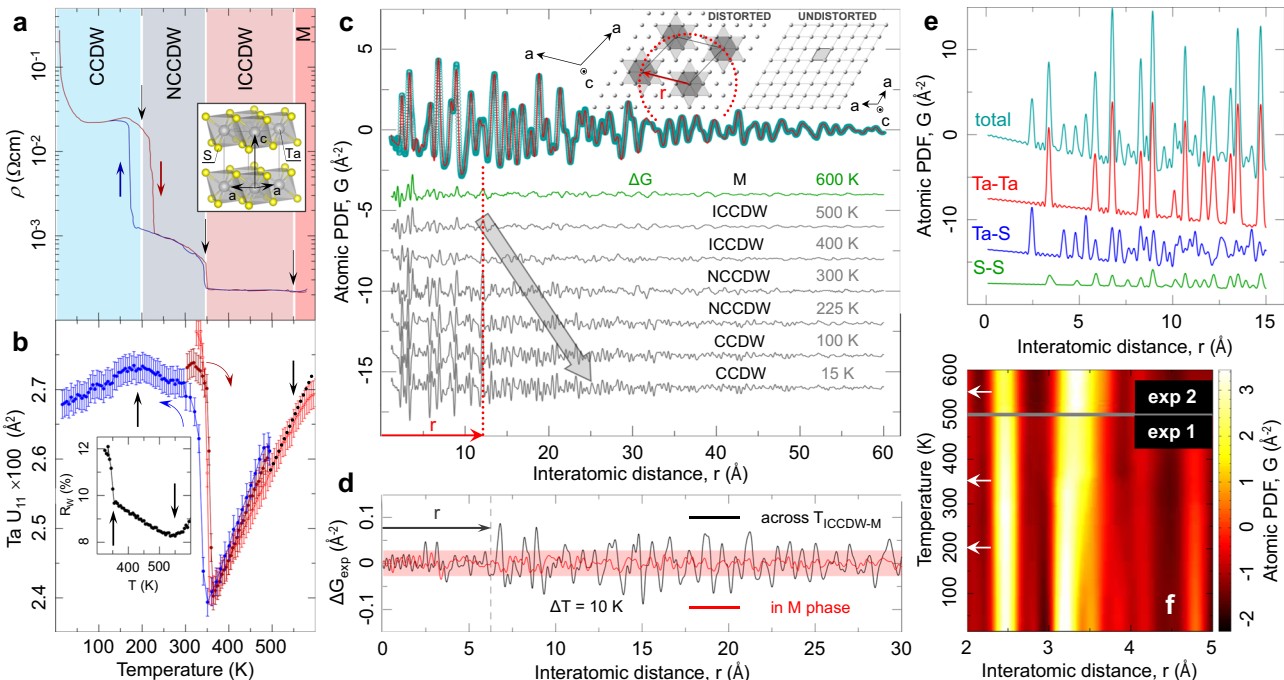

**Fig. 1 | Electronic transitions & lattice symmetry breaking in 1T-TaS₂.**
**a** Temperature dependent resistivity, adopted from Sipos et al.[9], highlighting electronic phases in 1T-TaS₂. The insert shows the undistorted P3̄m lattice structure. **b** Temperature evolution of in-plane ADP of Ta from P3̄m model fit to PDF data over 20–60 Å range. Red color indicates heating, blue color indicates cooling. Sloping black dotted line is a reference. Inset: Fit residual, $R_w$, implicates IC-M and IC-NC transitions. **c** P3̄m model fit to 600 K data. Green difference trace (offset for clarity) reveals short range deviations. Stack of difference curves for the same

model against data at different temperatures (as indicated) is shown underneath. Red arrow indicates a length-scale coarsely corresponding to the SoD polaron and the in-plane nearest neighbor inter-star center-to-center distance (inset). **d** Comparison of difference PDF between experimental data that are 10 K apart within metallic phase (T > 550 K) and across the IC-M transition, implicating existence of local distortions in the M phase over at least a radius r. **e** Calculated total PDF and partial pair-contributions in the P3̄m structure. **f** False color plot of PDF intensity versus temperature over a short r-range.

level[12,34]. While nominally metallic at high temperatures, it shows an incommensurate (IC) CDW (ICCDW) along the principal crystal axes already at high temperatures (between 550 and 350 K), with an associated Kohn anomaly at a wavevector $q_{IC} \simeq (0.283,0,0)$ in units of the reciprocal lattice vector $a_0^*$[36,37] which implies a strong coupling between electrons and the lattice. Below 350 K, the IC state breaks up into a patchy periodic network of commensurate islands separated by discommensurations[38–40]. Upon further cooling, this nearly commensurate (NC) CDW (NCCDW) state becomes fully commensurate (C) CDW (CCDW) after a first order transition at ∼ 180 K, with the appearance of a Mott gap in the charge excitation spectrum observed by single-particle tunneling[41] and angle-resolved photoemission[25,42]. In contrast, gapless spin relaxation in the range 50–200 K is attributed to a possible quantum spin liquid (QSL) phase, arising from the spin frustration on the triangular superlattice. While band structure calculations[3] suggest that the QSL phase may be suppressed by interlayer coupling[43], this cannot explain the marked difference of spin and charge excitation spectra[2]. A further intriguing feature is a discontinuity in the magnetic susceptibility and an anomalous peak in the spin-spin relaxation time concurrent with a crossover around ∼ 50 K of the spin-lattice relaxation time from a QSL-like, to a gapped T-dependence[44,45]. Near 50 K, a resistivity upturn from nearly T-independent resistivity to strongly T-dependent variable-range hopping behavior has been commonly reported, suggesting an onset of low-temperature charge localization[4,46]. The common picture of the 1T-TaS₂ lattice structure in the C state assumes simple distortions from the average lattice structure (trigonal P3̄m, Fig. 1a), whereby 12 Ta atoms are attracted symmetrically towards the central Ta atom that carries an extra electronic charge, forming a star of David (SoD) pattern (Fig. 1c). The SoDs are arranged in a $\sqrt{13} \times \sqrt{13}$ unit cell superlattice structure (P3̄ symmetry)[47]. At such an electron density of 1/13,

the dimensionless ratio $r_s$ of the Coulomb energy $V$ to the kinetic (Fermi) energy $E_K$ $r_s = \frac{V}{E_K} = \frac{e^2 m}{\hbar^2 \sqrt{n}} \simeq 70 \sim 100$, where $e$ is the elementary charge, $n$ is the (2D) density, and $m \sim 3$ is the effective electron mass[25]. Considering that the Wigner crystal stability limit is $r_s = 31 \sim 38$[23,48–50], 1T-TaS₂ is comfortably in the Wigner crystal regime, which means that electronic superlattice ordering on the basis of dominant Coulomb interactions and tell-tale lattice deformations around localized carriers may be anticipated. Room temperature extended X-ray absorption fine structure (EXAFS)[51], high-resolution transmission electron microscopy (HRTEM)[52–54] and X-ray structural measurements[44] indeed suggest the existence of symmetry-breaking atomic displacements, but studies over a wide range of ordering temperatures have not yet been performed.

Here we present the first systematic measurements of the local lattice structure that covers temperatures from 15 to 915 K. The data reveal local symmetry breaking from the established P3̄m symmetry at all measured temperatures (Fig. 1a, inset), and unambiguous evidence for individual polarons in the dilute high-temperature limit, revising our current notions about the origin of charge and spin ordering, with clear implications for other materials of current interest.

## Results

The local structure of 1T-TaS₂ is investigated using X-ray pair distribution function (PDF) analysis[55,56], giving picosecond-timescale structural snapshots for 15 K ≤ T ≤ 915 K. This encompasses the known electronically ordered states below 600 K portrayed by the resistivity vs. temperature plot shown in Fig. 1a[9], as well as irreversible polytype transformations above 600 K.

The anisotropic atomic displacement parameters, $U_{ij}$, of Ta atoms were retrieved from a fit to the established P3̄m structure over a range of 20–60 Å (Fig. 1b). For T < 600 K, the $U_{ij}$s, reveal an anomalously

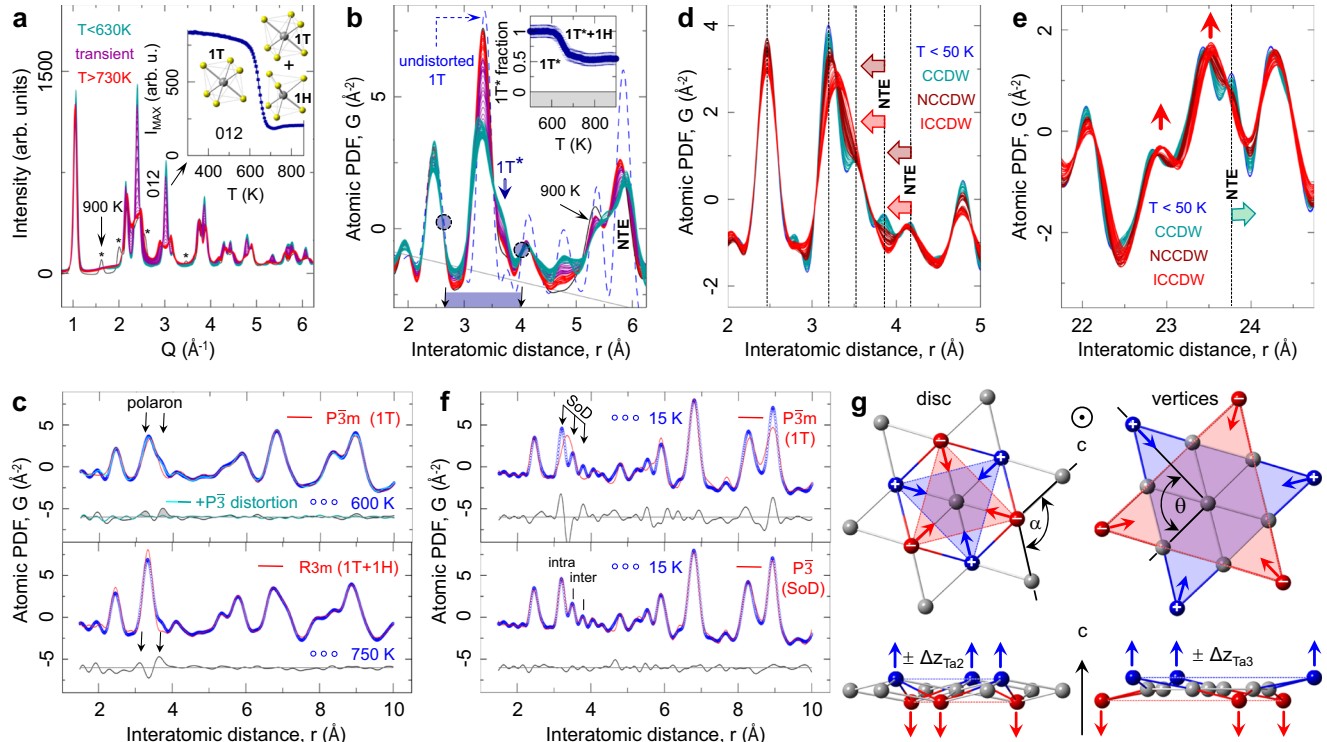

**Fig. 2 | Temperature progression of atomic structure in 1T-TaS₂. a** Stack of diffraction patterns revealing a polymorphic transformation on warming at ~630 K. Inset: intensity collapse of the (012) P$\bar{3}$m Bragg reflection during the restructuring from pure 1T to a hybrid 1T & 1H coordination. Further restructuring above 880 K results in additional Bragg reflections, marked by asterisk. **b** Corresponding PDFs over a na*rr*ow *r*-range. Data feature isosbestic points (circled) - nodes of temperature-invariant PDF intensity. Label 1 T* indicates presence of local distortions in the 1T-type layers. Vertical blue arrow (1T*) marks polaronic distortion intensity. Inset: Change of fraction of distorted 1T* environments, relative to 600 K, estimated over the shaded region on the abscissa. **c** Polaron signature, seen in the M phase (top), persists in the restacked polymorph structure (bottom), and is associated with the 1T-type layers. Adding distortions, described in

**g**, as nanoscale inclusions results in improved fit (cyan trace). In **d**, **e** a stack of PDF data on different length scales is shown over the 15–500 K range, color-coded according to the electronic phase they belong to. At high temperature longer length-scale peaks sharpen (indicated by vertical arrows) due to apparent average symmetry increase, despite expected broadening due to ADP increase at elevated temperature and in contrast to the shorter length-scale behavior. Features exhibiting negative thermal expansion (NTE) are indicated by block arrows. **f** At 15 K clear distortions associated with the star of David (SoD) are seen in the local structure, incompatible with the P$\bar{3}$m symmetry (top) but explained well (bottom) by a simple P$\bar{3}$ broken symmetry model, depicted in **g** with focus on Ta atoms. Colored arrows show the Ta displacements. Out-of-plane degrees of freedom (exaggerated in the illustration for clarity) provide for puckering distortions.

large in-plane component, $U_{11}$, approximately a factor of 2 larger than the component along the stacking axis, $U_{33}$ (e.g. at 500 K $U_{11}$ is 0.025 Å² whereas $U_{33}$ is 0.01 Å²). This is unusual for a layered system in which a larger out-of-plane component is expected to arise from interlayer disorder. At high T, in the metallic (M) phase, the large $U_{11}$ signifies an apparent intralayer 'disorder', unaccounted for by the P$\bar{3}$m structural model. At lower temperature, the P$\bar{3}$m model is inadequate at any temperature, revealed by the difference PDFs shown in Fig. 1c. The spatial extent and character of this symmetry breaking changes across the cascade of CDW transitions, tracking complex correlations in different electronic regimes. Intriguingly, as we describe below, nanoscale symmetry breaking is also evident in the polymorphs above 630 K (Fig. 2b, c).

As an aid to understanding, pair-specific contributions to the total PDF within the P$\bar{3}$m model are shown in Fig. 1e, where Ta-Ta pair contributions are dominant (see Methods). In the P$\bar{3}$m structure, the undistorted Ta sublattice features a single-valued nearest-neighbor Ta-Ta distance, seen as a sharp peak at ~3.4 Å, Fig. 1e. However, the data reveal a distinct additional shoulder, (e.g. Figs. 1e and 2b for 600 K), which is unexplained by the P$\bar{3}$m model (Figs. 1c and 2c (top)). At high temperature this distortion is most likely dynamic, as expected for symmetry-specific lattice fluctuations associated with carrier localization and polaron formation. Very specific distributions of Ta-Ta and Ta-S pairs in the 3.3-5.9 Å range are expected for SoD polaronic distortions[47]. In the C phase with a single layer $\sqrt{13} \times \sqrt{13}$ supercell P$\bar{3}$

model (Methods and Supplementary Table 1) the SoD polaron structure is adequately described (Fig. 2f)[47,57]. However, for all other electronic regimes it was necessary to add P$\bar{3}$m as a secondary phase to achieve good fits (see Fig. 3c, inset). The temperature dependence of the interatomic distances and bond angles, and deductions resulting from fits to the data are addressed sequentially below, starting from high temperatures, well above any known charge or spin ordered phase.

**Polymorphic regime**

Heating 1T-TaS₂ above 600 K results in irreversible polymorphic transformations altering average symmetry and stacking[26,51], where nominally 50% of octahedrally coordinated 1 T layers convert into trigonal prismatically coordinated 1 H layers[35,58] (Fig. 2a inset). This is accompanied by polymorph-specific stacking of the two layer-types[4,45] and changes of the local Ta environment in the prismatic layers. Two such transformations are seen in the PDF data (Fig. 2b), at 630 and 880 K (see Supplementary Figure 1 for more details). Remarkably, a shoulder-signal ( ~ 3.5 Å, Fig. 2b) associated with local lattice distortions (polarons) is observable for 730 K < T < 915 K. The distorted fraction does not vanish in the polymorphs but instead drops to half its value observed in the 1 T regime (T < 630 K), with no significant reduction through the second transformation (Fig. 2b). Notably, PDF data above 730 K conform to the 6R-TaS₂-like model (R3m symmetry) featuring equally abundant alternating 1T and 1H layers (Fig. 2c,

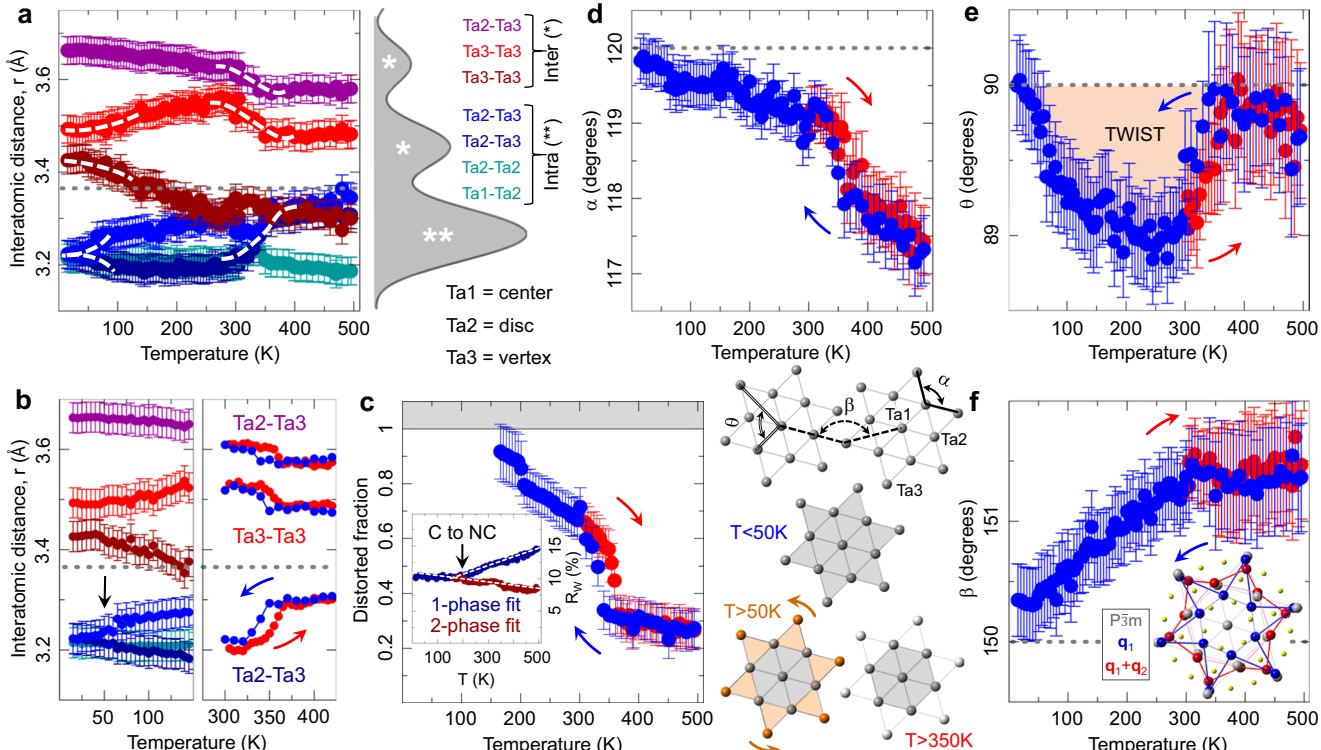

**Fig. 3 | Local structure quantification from distorted local SoD model. a** Local Ta-Ta distances from a P3̄ model fits to PDF data, obtained on cooling, over 1 nm length-scale in 15–500 K range. For reference, dashed horizontal gray line marks the undistorted P3̄m value at 300 K. Intra- and inter-star distances are color coded as indicated in the legend. **b** Closeup of C phase behavior (left) and hysteretic response across the IC-NC transition (right). **c** In NC and IC phases secondary undistorted P3̄m phase was necessary to reduce fit residual, $R_w$ (inset). Distorted phase fraction, displaying thermal hysteresis consistent with one seen in resistivity, is calculated from atomic weight-based phase content obtained from the fits. **d**–**f** show selected intra-SoD and inter-SoD bond angles, as indicated in the sketch.

In the IC phase the SoD motif is heavily distorted, with local distortions resembling discs. Upon entering the NC phase, the SoD distortion becomes better differentiated, with stars exhibit twists, and with interatomic angles beginning to evolve towards ideal values. In the C phase twists start to diminish. Below ~50 K a sharp regularization of stars is observed, with intra-star Ta-Ta distances becoming equal, intra-star angles approaching ideal values, and no twisting. Colored arrows in **b**–**f** indicate data collection on warming (red) and on cooling (blue) cycles. Inset to **f**: displacements of the Ta atoms for the $q_1$ (blue) and $q_1 + q_2$ (red) modes from symmetry analysis at 15 K, compared to the parent P3̄m phase (gray).

bottom)[59]. The model features undistorted 1T and 1H layers, but a clear distortion signal, albeit weaker, such as is seen at 600 K, persists in the fit differential, implying the existence of polaron-specific deformations in the 1T layers well above 600 K (Fig. 2c, top), with ~10 wt% distorted fraction (see below).

## High-temperature metallic regime

The data at 600 K show multiple components at the position where the P3̄m model predicts a single Ta-Ta peak (Fig. 1c and red trace fit in Fig. 2c, $R_w = 15\%$). The distortion spans ~5 Å with weaker signal extending up to ~12 Å, depicted by the difference PDFs. Such signatures, also seen in the IC phase, are weaker than in the NC and C regimes (Fig. 1c). However, their similarity and continuity imply a common origin and unambiguously reveal the presence of high temperature electron localization, detectable by charge-lattice coupling[60,61]. Despite similar P3̄m model misfits shown in Fig. 1c, explicit data differences, Fig. 1d, reveal subtle changes across the M-IC transition for $r > 5$ Å. The electron localization results in incomplete breakup of the P3̄m matrix, but without the formation of the full SoD motif. The light red profile in Fig. 2d and the intensity band in Fig. 1e at ~3.3 Å remain broad, consistent with incoherent distortions from weakly developed fluctuating charge correlations. In this diluted limit local distortions were approximated as P3̄ nanoscale inclusions, Fig. 2g, within the undistorted P3̄m matrix. This model (cyan traces in Fig. 2c (top), $R_w = 8\%$) accounts for the misfit, resulting in ~20 wt% of P3̄ distortions involving heavily puckered hexagonal Ta discs surrounding

central tantalum. (This is consistent with ~10 wt% of locally distorted environments in the 1T-layers in the polymorphic regime above 630 K discussed in the previous section).

## Incommensurate regime

The growth of charge correlations and structural coherence of associated distortions results in their signature extending over a longer length-scale (Fig. 1c). Stacks of experimental PDFs for 15-500 K (Fig. 2d, e) show that short length-scale peaks ($r < 5$ Å) sharpen gradually on cooling. However, the data still do not show a clear SoD motif (Fig. 2d). The data at ~3.5 Å are broad and unresolved, and at ~3.75 Å are rather featureless. Conversely, peaks at higher distances show the opposite temperature trend. In the light red stack, Fig. 2e, the signal at ~23.5 Å is the sharpest just below the M-IC transition, portraying high symmetry, and systematically broadens with intensity reduction as temperature decreases, evidencing growth of broken-symmetry correlations. The change in slope of Te $U_{11}(T)$ at 550 K (Fig. 1b) indicates an increased localized charge density. Model-derived Ta-Ta distances for $T > 350$ K shown in Fig. 3a portray substantially distorted hexagonal SoD cluster cores comprised of short Ta-Ta contacts (Fig. 3d, f). The distorted fraction increases to ~30 wt% compared to the M state (Fig. 3c). The PDF peaks which are sensitive to interlayer stacking are very broad, as shown in Fig. 4a, c, implying inhomogeneous stacking. The distribution centroids of the closest interlayer separation, Fig. 4a, are shifted to the left, well below 5.9 Å, indicating local compression of layers which are, on average, closer together than they would be in the undistorted

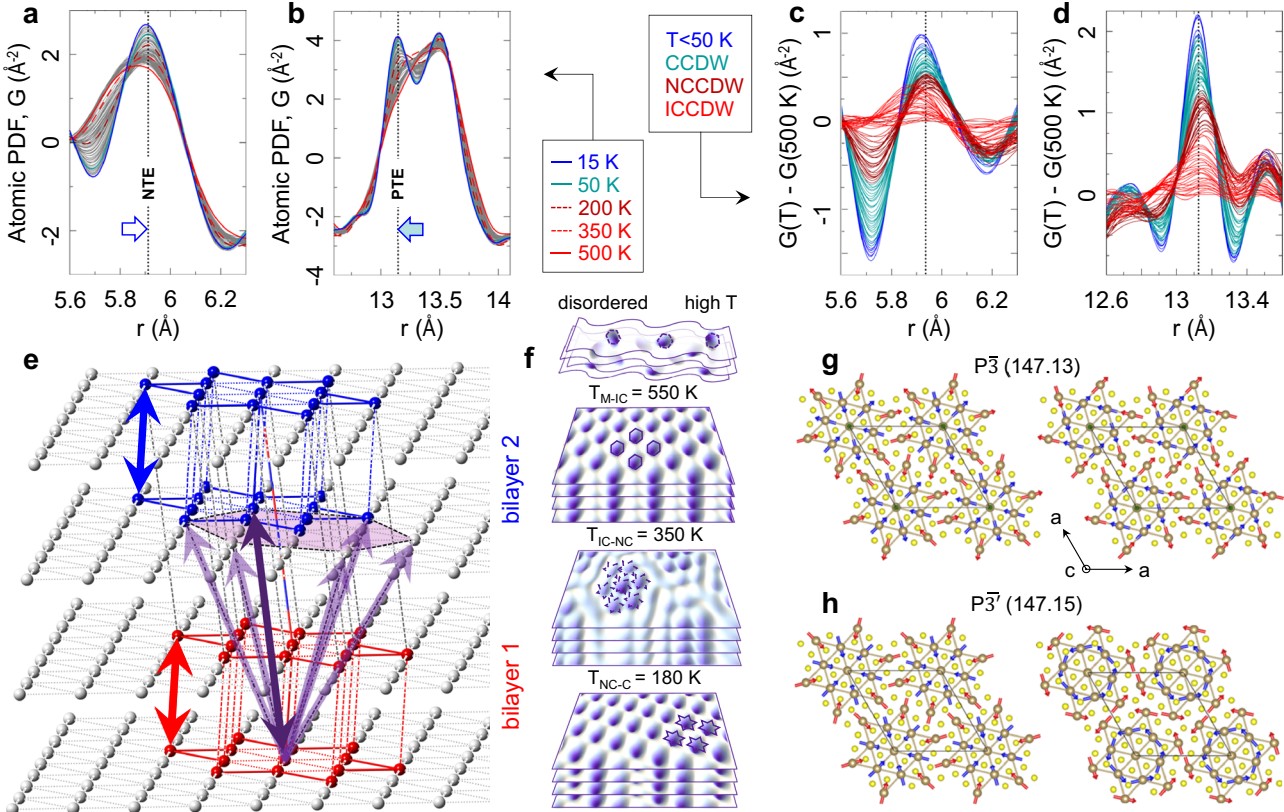

**Fig. 4 | Model independent considerations of stacking correlations.**
**a** Temperature stack of experimental PDFs around ~5.9 Å peak containing *intra*-bilayer correlations of SoDs, indicated in **e** by blue and red double arrows.
**b** Temperature stack of PDFs around ~13.1 Å peak depicting *inter*-bilayer correlations of SoDs, indicated in **e** by purple double arrows. The contribution assignment is made based on features in P3̄m model, Fig. 1e. There, the 5.9 Å feature is the c-axis lattice repeat distance involving strong signal from the nearest neighbor interlayer Ta-Ta correlations, whereas the 13.1 Å feature is well isolated and dominated by the next nearest neighbor off-axis interlayer Ta-Ta correlations. In **c**, **d** temperature stacks of differential PDFs are shown for data in **a**, **b**, respectively, with differentials calculated using 500 K reference. Vertical dashed lines mark positions of apparent

maxima in the C phase. **e** Sketch of two bilayers highlighting correlations discussed in text. Inter-bilayer SoDs are separated by ~13.1 Å at low temperature, a peak forming on cooling from a broad distribution at high temperature. This separation results from a SoD-shift by approximately one P3̄m lattice spacing in each P3̄m lattice direction. Note 6 possible choices of relative positioning of nearest bilayers, indicated by the purple shaded hexagon. **f** Schematics of distortions across different electronic phases in 1T-TaS$_2$. **g**, **h** magnetic moments consistent with the distortions of the SoDs as determined by the PDF analysis for two different settings. The arrows at the center Ta1 atom pointing along the z axis are shown in green (only in P3̄m).

1T structure. The observation of negative thermal expansion jumps along the stacking axis is relevant in this context[62] (see Supplementary Figs. 2 and 3 for discussion).

**Nearly commensurate regime**

Significant local structure changes occur at the IC-NC transition. The distortion fingerprint, Fig. 1c, grows dramatically and PDF intensities vividly redistribute below 5 Å (Figs. 1f and 2d). The Ta-Ta peak at 3.25 Å, which is broad in the IC phase, splits into an incompletely resolved triplet of unequal intensity peaks at ~3.15 Å, ~3.3 Å, and ~3.8 Å, marking the formation of well-defined SoD clusters. The feature at ~3.15 Å corresponds to the Ta$_{13}$ SoD intra-cluster configuration, whereas the other two features depict the inter-cluster correlations (Fig. 3). Residual intensity at ~3.4 Å is associated with discommensuration regions of approximately P3̄m character, separating hexagonal domains of well-ordered SoD clusters within the Ta layers[63,64], well observable by STM. Multiphase modeling with coexisting P3̄ and P3̄m phases confirms this picture (Fig. 3c inset, see Methods). As SoD clusters emerge, the distortion magnitude increases (Fig. 3a, b), consistent with increasingly resolved signal seen for 3 Å < r < 4 Å shown in Fig. 2d. Importantly, the star vertices exhibit sudden asymmetric contraction, accompanied by substantial puckering (Fig. 3d, f) and SoD twisting (Fig. 3e). The distorted P3̄ phase becomes the dominant fraction below 350 K, at

~60 wt% (Fig. 3c). This is consistent with the patchy nature of the discommensurate NCCDW phase. Narrow thermal hysteresis, shown in Fig. 3b–e, reflects the 1st order character of the IC-NC transition. As SoD local order is established in the NC state, they simultaneously form well defined bilayer correlations, Fig. 4a, c: The SoDs in adjacent layers couple and align along the c-axis. In contrast, inter-bilayer correlations are still fuzzy: different bilayers exhibit appreciable stacking disorder, evident from inter-bilayer sensitive peak, Fig. 4b, d, which probes off-axis correlations whose distribution is still considerably broad in the NC state.

**Commensurate regime**

At the NC-C transition, the PDF intensity at ~3.4 Å diminishes, implying the gradual disappearance of the ordered domain walls, while the intra-SoD-cluster peak sharpens (Fig. 2d), consistent with increased SoD density and additional charge localization. Similar changes are observed in the far-r limit (Fig. 2e), as a homogeneous crystal structure forms in the commensurate state. Closer inspection of the data unveils an additional restructuring below ~50 K, embodied in subtle peak shifts and sharpening observed in the dark blue traces in Figs. 2d and 4a. Modeling shows that the distribution of Ta-Ta distances depicting intra- and inter-SoD Ta-Ta contacts changes character in the C phase, with tightly bound SoDs structurally further regularizing below 50 K

(Fig. 3a, b). The intra-SoD angle θ reduces, approaching 120° at base temperature, Fig. 3e. The intra-star twist angle α increases, approaching 90° below 50 K (Fig. 3d), while the local inter-star angle β reaches 150° as SoDs align and form fully commensurate order (Fig. 3f) and the undistorted phase diminishes, consistent with vanishing domain walls (Fig. 3c). Simultaneously, well defined inter-bilayer correlations develop (Fig. 4b, d). The most dramatic changes are observed at -13.1 Å where PDF intensity redistributes and sharpens, possibly indicating an offset of pairs of SoDs, going from one bilayer to another, by one P$\bar{3}$m lattice spacing along both planar lattice vectors, Fig. 4e. Notably, there are 6 choices of such offsets, consistent with 13× stacking period and helical stacking arrangement suggested in some works[44]. Further qualitative changes are seen below 50 K in the PDF peak which includes the nearest neighbor intra-bilayer stacking peak just below 6 Å, whose intensity distinctly and abruptly shifts to lower distance (Fig. 4a, c). In contrast, the inter-bilayer peak at -13.1 Å sharpens, but does not shift (Fig. 4b, d). This could imply enhanced intra-bilayer coupling, with neighboring bilayers presumably adjusting to preserve the inter-bilayer spacings[36].

## Symmetry of the order parameter

On the basis of a symmetry analysis of the observed PDF displacements we can identify the symmetry of order parameters which transform as the $A_{2u}$ and $E_u$ irreducible representations in the parent P$\bar{3}$m structure space group. The displacements are described by two modes with wavevectors of the form $q_1 = (a,b,0)$ and $q_2 = (-b,a+b,0)$ at 60° to $q_1$. In the C phase, $q_1 = \frac{3}{13}a^* + \frac{1}{13}b^*$, and $q_2 = -\frac{1}{13}a^* + \frac{4}{13}b^*$, where $a^*$ and $b^*$ are the reciprocal lattice vectors of the undistorted lattice. It is possible to decompose the atomic displacements of the Ta atoms at any temperature in terms of the independent amplitudes of the $A_{2u}$ and $E_u$ modes described by $q_1$ and $q_1 + q_2$. The displacements at 15 K are shown in the insert to Fig. 3f. In the NC phase at room temperature[44], $a_{NC} = 0.2448(2)$ and $b_{NC} = 0.0681(2)$, which is close to the C values ($a_C = 3/13$ and $b_C = 1/13$), while in the IC phase, $a_{IC} = 0.283(2)$, and b = 0, where $a_{IC}$ is close to 2/7 = 0.2857. Note that our PDF analysis does not reveal any information about long range order, and $q_1$ and $q_2$ should not be interpreted to imply its existence. Rather, they describe the symmetry of the lattice distortions associated with the twists and displacements of the Ta atoms in the distorted SoD polaron structures. (See Supplementary Fig. 4).

## Discussion

The local structure data in 1T-TaS$_2$ reveals symmetry-specific local structural fingerprints of polaron formation whose ordering evolves with temperature as shown schematically in Fig. 4f. Surprisingly, the polaron fingerprints are already identifiable in the polymorphic 6R-TaS$_2$ regime (where individual 1T-monolayers appear between 1H layers), in the metallic phase (> 600 K), then successively through the IC, NC and C ordering transitions, and eventually in the non-QSL-like regime below 50 K, where—remarkably—SoD symmetry is restored in a layer-dimerized undistorted in-plane SoD structure. An important fundamental question arises in the context of conventional CDW viewpoint regarding the lattice distortions in the high-temperature M phase: Should the deformations be discussed in terms of a uniform reduced modulation amplitude in the IC-like phase, or in terms of dynamical polaron fluctuations? Fig. 1d indicates the range of the structural correlations. The black trace shows a difference across the IC-M transition, while the red trace is a difference corresponding to two PDFs in the same (M) phase separated by the same $\Delta T$. This comparison shows that dominant changes across IC-M occur at $r > 5$ Å, with smaller changes for $r < 5$ Å (Supplementary Fig. 5). This implies that at the M to IC transition both the number of distorted sites and the extent of IC spatial correlations grow. The M phase thus looks like a sparse polaron gas, whereas the IC phase is a sparse polaron crystal (incommensurate with the lattice). We can now consider the three

possible scenarios for the IC melting transition based on the PDF data: In the first one, in the M phase, the long-range IC-CDW becomes dynamic, with amplitude and phase fluctuations on a timescale faster than the lattice can respond. In the second scenario, all relevant charges become fully itinerant on all length scales in the M phase without lattice distortions. In the third, polaronic, scenario the IC-CDW breaks up in the M regime but the lattice follows the localized charge fluctuations, which is equivalent to thermally activated polaron hopping. In all three scenarios, we would see a disappearance of the reflections at the IC modulation wave vector on going from IC to M in Bragg reflections, with the third scenario leaving some diffuse scattering in the M phase. In the PDF data, the first and second scenario would result in no characteristic polaronic distortions in the M phase. The fact that we *do* see them implies that only the polaron fluctuation scenario is relevant in 1T-TaS$_2$. The persistence of polarons in the polymorphic regime (Fig. 2b, c) and the change from 20 wt% to 10 wt% on the transition from a uniform 1T to the 6R multilayer structure confirms this notion and implies that polarons exist only in individual 1T monolayers. These may be expected to appear also in free-standing monolayers or monolayers on substrates, provided that strain and interaction with the substrate do not interfere.

## Magnetic ordering

It is interesting to consider the symmetry of the magnetic polaron structures that are compatible with the two possible in-plane magnetic space groups P$\bar{3}$ (#147.13) and P$\bar{3}$′ (147.15). Two (ferro)magnetic P$\bar{3}$m subgroups can be derived from the analysis of the symmetric irreducible representation at the Γ point of the trigonal Brillouin zone of the 1T stack. For P$\bar{3}$m, inversion does not flip the magnetic moment, allowing a magnetic moment component along c for Ta1 and no constraints for the magnetic moments of the Ta2 and Ta3 atoms. In contrast, for the P$\bar{3}$′m magnetic space group, inversion reverses time and thus flips the magnetic moment, the magnetic moments of the Ta atoms of the same kind are not flipped by the symmetry operators. By choosing different components for Ta2 and Ta3 magnetic moments it is possible to create different motifs maintaining the same symmetry. Figure 4g, h show the motifs compatible with the symmetry operations of the two magnetic groups for in-plane chiral and non-chiral arrangements of magnetic moments respectively. An in-plane QSL-like state is consistent with fluctuating in-plane moments, and c-axis antiparallel moments in the P$\bar{3}$′ magnetic group, consistent with dimerized stacking of CDW orders in the C state.

A remarkable feature of the data is that local in-plane mirror symmetries appear to be restored below 50 K, with $\alpha \simeq 120°$ and $\beta \simeq 150°$. The data below 50 K are strongly suggestive of dimerization along the c axis, consistent with inter-plane spin singlet formation and spin gap formation, that could explain the apparent cross-over from a $\sim T^2$ QSL-like spin relaxation above 50 K to a gapped, exponential T dependence below this temperature[2]. A plausible origin of the apparent symmetry restoration at low temperature comes from the disappearance of domain walls at low temperature. In-plane domain walls have been observed to persist well below the NC-C transition temperature, which topologically prevent interlayer stacking and spin dimerization. As long as domain walls exist, un-dimerized layers support an intra-layer QSL composed of spin structures shown in Fig. 4g, h, while regions with dimerized lattice layers are expected to exhibit inter-layer spin pairing[3]. The restoration of SoD symmetry at lowest temperatures may thus be considered to be associated with the disappearance of DWs. A slightly different situation occurs in the metastable 'hidden' domain state[65], where a QSL is allowed to form within domains whenever inter-layer stacking is indirect.

The polaron crystallization paradigm[66] emerging on the basis of these data has the benefit of providing a common framework for the understanding the polaronic physics in both the equilibrium and metastable phases of 1T-TaS$_2$ and numerous related materials that

display similar charge and spin ordering[23]. The local structure symmetry analysis is particularly relevant for revealing the origin of the still poorly understood spin polaron structure and reconciles the observation of QSL-like behavior with theoretical band structure modelling, which commonly suggests a spin-paired dimerized ground state in a perfect crystal[3]. Finally, we note that it was shown theoretically that Wigner-crystal ordering is not limited to systems with a half-filled electronic band, but may be also present at other fillings[21,67], which suggest that the crystallization phenomenon reported here may be more general than previously thought.

## Methods

### Crystal growth and analysis
The 1T-TaS$_2$ samples were grown using iodine vapor transport reaction, grown at 850 °C, and quenched to room temperature from the growth temperature. Single crystal XRD shows a pure single 1T phase is retained after the quench, with Ta:S composition determined by EDS to be 33 : 66 ± 1at%.

### Synchrotron X-ray atomic pair distribution function (PDF) measurements
Temperature dependent X-ray total scattering data were collected at 28-ID-1 beamline of the National Synchrotron Light Source II (NSLS II) at Brookhaven National Laboratory. Finely ground powders of 1T-TaS$_2$ were sealed in 1 mm (outer diameter) polyimide capillary (referred to as experiment 1 or exp 1) and 1.5 mm (outer diameter) quartz capillary (experiment 2 or exp 2) within a glovebox under light vacuum. Measurements were carried out in capillary transmission geometry using a 2D PerkinElmer amorphous silicon area detector (2048 × 2048 pixels with 200 μm$^2$ pixel size) placed ~204 mm downstream of the sample. The setup utilized a monochromatic X-ray beam with 74.5 keV energy (λ = 0.1665 Å). Sample temperature control was achieved using a Cryo Industries of America cryostat (exp 1, 15 K ≤ T ≤ 500 K) and a FMB Oxford Hot Air Blower model GSB1300 (exp 2, 300 K ≤ T ≤ 915 K). Data for each experiment were collected in 5 K steps using 5 K/min temperature ramp, 2 min thermalization, and 2 minutes data collection at each temperature. In exp1 data in temperature range 15 K ≤ T ≤ 300 K were collected on cooling, while these in temperature range 300 K ≤ T ≤ 500 K were collected on warming and cooling. In exp 2 data were collected on warming.

### PDF data processing and analysis
Calibrations of the experimental geometry, momentum transfer range, and detector orientation were carried out by utilizing nickel standard measurements performed under the same conditions. Appropriate masking of the beam-stop shadow, inactive and outlier pixels, and subsequent azimuthal integration of the 2D images to obtain 1D diffraction patterns of intensity versus Q data were done using pyFAI software package[68,69] Standardized corrections to the data for experimental effects to obtain the reduced total scattering structure function, F(Q), and the subsequent sine Fourier transforms to obtain experimental PDFs, G(r), with $Q_{max} = 25$ Å$^{-1}$ (exp 1) and $Q_{max} = 20$ Å$^{-1}$ (exp 2) were carried out using the PDFgetX3 program within the xPDFsuite software package[70] The PDF analysis was carried out using the PDFgui[71] modeling platform. The PDF, derived from powder diffraction-based Bragg and diffuse scattering data collected over a broad range of momentum transfer, describes direction-averaged distribution of atom pairs in a material as a function of interatomic distance, r, and provides structural information across different length scales. Uncertainties in reported model-derived parameters were estimated from their maximum variations observed by changing either the r-range of modelling (Figs. 1b, 3a–f) or integration (Fig. 2b, inset) by 10%. A double of the maximal values of such uncertainties was adopted as the error bars shown. Due to large X-ray scattering contrast (Z(Ta) =73, Z(S) = 16) the dominant contribution to total PDF (Fig. 1e) originates from Ta-Ta pairs (red trace), followed by the Ta-S pairs' contribution (blue trace). The S-S (green trace) contribution is about 20 times weaker than the Ta-Ta contribution and 4 times weaker than that of Ta-S pairs.

### Estimate of distorted fraction and structure modeling in polymorphic regime
By using a calculated PDF based on undistorted P3̄m model as a reference, we form difference PDFs for all T > 600 K data. Such differences are then integrated over a 2.6–4 Å range (highlighted region on abscissa in Fig. 2b) and normalized to the 600 K value to quantify the relative change with temperature of the fraction of distorted sites across the polymorphic transformations, as shown in inset to Fig. 2b (for illustration see Supplementary Fig. 1). Above 730 K the short-range PDF data can be explained by a 6R-TaS$_2$-like model (R3m symmetry) featuring alternating 1T and 1H layers in equal abundance, Fig. 2c (bottom). Although the exact stacking and long-range character are likely more complex, as the model does not fully explain the data over longer length-scales, local analysis is rather robust. Importantly, the 6R-TaS$_2$ model features undistorted 1T & 1 H layers, hence not accounting for local distortions responsible for nontrivial nearest neighbor Ta-Ta distance distribution.

### Local structure model with P3̄ symmetry
To map out temperature evolution of local interatomic distances, first data fits were carried out over 1 nm range and utilized a simple single S-Ta-S layer 1T-TaS$_2$ model conforming to P3̄ symmetry. The model includes 19 independent structure parameters, constrained by symmetry to describe fractional coordinates of Ta (6 parameters) and S (13 parameters) atoms in the unit cell, two lattice parameters, two thermal parameters, and an overall scale factor. This explains the data features in the C state very well. The refinement range was established as a compromise between the model over-parametrization (data window too narrow) and the model deficiency (data window too wide) regimes. The 1 nm fitting range encompasses intra-bilayer correlations but excludes inter-bilayer correlations, which were subsequently explored in a model-independent manner as described below. (See Supplementary Table 1, Supplementary Figs. 6 and 7, and Section 2 for additional model details.)

### Multiphase treatment
The fits using only the P3̄ phase in the NC regime were noticeably worse compared to these in the C state, with observable increase of fit residual, as shown in inset to Fig. 3c. This is consistent with the presence of discommensuration domain walls and lower SoD density in the NC state whose atomic structure is less distorted and not accounted for in our simple distorted P3̄ model. We approximated domain wall contribution by adding an undistorted P3̄m structure as a secondary (minority) phase to account for local environment of the discommensuration regions. We utilized this approach to fit the PDF data in the IC and M regimes as well, where the distorted and undistorted phases swap roles and the distorted phase becomes a minority phase.

### Assignment and consideration of inter-layer PDF peaks
The inter-layer PDF peaks considered in this study were assigned based on undistorted P3̄m model as reported for 300 K and its supercell variants. In this, interlayer spacings along the stacking direction are defined by multiples of c-axis lattice parameter, including ~5.9 Å which corresponds to intra-bilayer separation. The ~13.1 Å peak in undistorted P3̄m structure corresponds to a well-defined off-axis Ta–Ta contribution connecting two Ta atoms that are 2 layers apart. These Ta atoms belong to next nearest neighbor Ta-layers and are offset within planes away from the stacking axis. In the high temperature regime, if structural distortions are incoherent, this peak in the data would be

smeared, as is observed. As order establishes, re-appearance (with respect to putative P$\bar{3}$m structure that never materializes locally) of a sharp signal at ~13.1 Å implies re-emergence of structural coherence along this interatomic vector. The signal strength likely implies alignment of SoDs belonging to neighboring bilayers along this direction.

## Data availability

All of the data supporting the conclusions are available within the article and the Supplementary Information. Additional data are available from the corresponding authors upon request.

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

## Acknowledgements

We wish to acknowledge Igor Vaskivskyi, Jaka Vodeb and Viktor Kabanov for valuable discussions. Work at Brookhaven National Laboratory is supported by the Office of Basic Energy Sciences, Materials Sciences, and Engineering Division, U.S. Department of Energy (DOE) under Contract No. DE-SC0012704. This research used beamline 28-ID-1 of the National Synchrotron Light Source II, a U.S. Department of Energy (DOE) Office of Science User Facility operated for the DOE Office of Science by Brookhaven National Laboratory under Contract No. DE-SC0012704. DM and PS wish to acknowledge funding from ARRS. The work at the Jozef Stefan Institute was supported by the Slovenian Research Agency (P1-0040).

## Author contributions

E.B. and M.A. designed and conducted the X-ray experiments, E.B. performed PDF analysis, G.B. and S.C. performed group theoretical analysis. D.M. and E.B. conceptually proposed the experiments. E.B. and D.M. wrote the paper. P.S. synthesized and characterized the crystals.

## Competing interests

The authors declare no competing interests in this work.
