## [Peer Review File · Nature Communications]

REVIEWER COMMENTS

Reviewer #1 (Remarks to the Author):

Here Bozin and colleagues report a systematic series of structural data, namely angle-integrated pair correlation curves, from very high to low temperature in 1T-TaS₂.

They describe the various stages of the onset of pair correlations, most prominently Ta-Ta pairs, associated with the star-of-David motif that makes up the ground state C-CDW. One of the most interesting and surprising aspects of this paper is that the Authors are able to show that the detailed structure among Ta ions in the C-CDW phase continues to develop in subtle ways even below the transition temperature of 180 K. This is said to be partly due to extinction of domain walls (it is quite hard to imagine how this would happen, microscopically, but the evidence presented is interesting). Also very interesting and important, the Authors show evidence of the bilayer-correlated out-of-plane stacking, and show that it strengthens as temperature decreases through the C-CDW phase. This is argued to constitute evidence for the increasing exclusion of any quantum spin liquid approaching the ground state, due to increasing fraction of dimerized stars-of-David forming a valence bond solid.

The Authors also examine behavior in the very high temperature case in which half the 1T polytype transforms into 1H. The fraction of star-of-David clusters reduces indicating that the polarons reside in only the 1T layers.

The Authors' main emphasis is that even at high temperature in the metal phase there are deviations, as seen in the residual after a model fitting procedure, from the expected pair correlation curves for the high-symmetry P3-m structure that is usually thought to describe this phase of 1T-TaS₂. And this is argued to constitute evidence for a gas of polarons that will sequentially liquify and freeze as temperature is decreased.

The manuscript is structured in a straightforward way and is fairly well written. There is perhaps a bit too much unnecessary information in the rather dense and long introduction, and as a result it may not be very accessible for non-experts coming into the field. The references are, as far as I can judge, comprehensive and suitable. The data are convincing and I cannot find much that should be improved.

I think there are numerous interesting and impactful observations reported in this manuscript. It is certain to contribute to the discussion of inter-layer effects in 1T-TaS₂ (e.g. further evidence for the out-of-plane bilayer correlations), and also to the discussion of a putative quantum spin liquid. It also advances a compelling microscopic picture, that is easy to comprehend, for the formation of the various charge ordered phases of 1T-TaS₂ from the crystallization of a polaron fluid. And there are many other small points of interest. In light of the above I am happy to recommend the paper for acceptance without revision.

Reviewer #2 (Remarks to the Author):

The paper presents results from a study on the temperature evolution of the local atomic structure of the transition metal dichalcogenide 1T-TaS₂ using the atomic pair distribution function (PDF) technique. The authors conclude that the charge ordering exhibited by 1T-TaS₂ at low temperatures is driven by polaron crystallization and not Fermi surface nesting or conventional electron-phonon coupling. While the experimental PDF data are impressive, their interpretation is inconsistent, casting doubt on the main conclusions of the paper.

To be more specific, for all phases exhibited by 1T-TaS₂, the authors use a structure model based on a single TaS₂ atomic layer, often aided by an inclusion of a second phase. The pronounced lattice distortions in 1T-TaS₂, as reflected by the atomic PDFs, are modeled by unphysically large, out-of-plane displacements of Ta atoms alone (Fig. 2g) while the modulation of S atomic planes sandwiching the Ta planes, i.e., the positional disorder of S atoms, is disregarded. The very significant contribution of Ta-S pair correlations (together with the non-negligible contribution of S-S correlations !) to fine features of the experimental PDF data, which could be affected by S disorder considerably, is not accounted for either.

The overall impression is that the authors tend to employ overly simplistic atomic configurations to characterize the complex structure of 1T-TaS₂ and evoke a phase segregation scenario to explain the resulting large discrepancies between the model computed and experimental PDF data. This leaves a lot of room for misinterpretation of the PDF data. The authors need to explore more realistic model atomic configurations that explain the PDF data well and then try to explain the remaining small discrepancy between the model and computed PDF data, if any, and not the other way around.

Furthermore, the authors tend to attribute evolution of selected PDF features (Fig. 4a,b, Suppl. Fig 3d and Suppl. Fig. 5b,c) to evolution of assumed structural features of 1T-TaS₂ without presenting evidence that their assumptions are correct. The authors also make conclusions based on miniscule changes in the PDF data (e.g. Suppl. Fig. 5b,c) that are comparable in magnitude to Fourier

transformation ripples and data noise (compare with the low-r part of the PDF data in Fig. 2c,f and Suppl. Fig. 5a). The authors should convince the readers that such changes are real structural features of 1T-TaS₂ and not an experimental artifact.

In conclusion, unless my criticism is addressed properly, I may not recommend the paper for publication.

Response to referee #2

The paper presents results from a study on the temperature evolution of the local atomic structure of the transition metal dichalcogenide 1T-TaS₂ using the atomic pair distribution function (PDF) technique. The authors conclude that the charge ordering exhibited by 1T-TaS₂ at low temperatures is driven by polaron crystallization and not Fermi surface nesting or conventional electron-phonon coupling. While the experimental PDF data are impressive, their interpretation is inconsistent, casting doubt on the main conclusions of the paper.

Author's response: We thank the referee for praising the data. We very much appreciate the points raised by the referee, which we realize in large part arise from an insufficiently detailed description of our model in the submitted manuscript. We regret the confusion this has caused. In the revised manuscript and supplement we have incorporated a much more detailed description for the model, which we believe removes possible ambiguity and misinterpretation, and directly addresses the issues that the referee has raised. We note that the main conclusions are largely independent of the detailed model. We believe that the revised manuscript and supplement address these aspects, and thank the referee for their careful assessment which contributed to strengthening the manuscript.

To be more specific, for all phases exhibited by 1T-TaS₂, the authors use a structure model based on a single TaS₂ atomic layer, often aided by an inclusion of a second phase.

Author's response: We have added missing details to clarify the models that we employed in our analysis. Our basic model involves a S-Ta-S slab, but some features (particularly the sharp PDF Ta peaks) cannot be accounted for without taking into account nearest neighbor slabs as well. The fits include all atoms within 1 nm radius, which includes nearest-neighbor layers, including the possibility of correlated bilayers, which previous literature shows to be the most common repeat unit along the c axis. We use a reasonable compromise between the over-parametrization (model too complex for the data range) and underfitting (model inadequacy). A justification of the model, now described in the Methods section, is presented in the revised supplement.

The second 'phase', or more accurately, the effect of discommensurations, is included in the temperature range where evidence from previous literature shows that it is present (in the NC phase). Not doing so would be hard to justify. The fitting confirms its presence, and justifies its inclusion in the model. The two phases represent two distinct local environments present in the NC and higher temperature phases.

The pronounced lattice distortions in 1T-TaS₂, as reflected by the atomic PDFs, are modeled by unphysically large, out-of-plane displacements of Ta atoms alone (Fig. 2g) while the modulation of S atomic planes sandwiching the Ta planes, i.e., the positional disorder of S atoms, is disregarded.

Author's response: This issue also arises from our incomplete model description, and we apologize for the confusion this has caused. We note that the displacements in Fig. 2g are sketched to emphasize the degrees of freedom of the model. The displacements were magnified 10x for emphasis, but this was not pointed out in the text, for which we apologize. We now state this explicitly in the figure caption, and explicitly state that the sketch only depicts the Ta aspect. Since no other data on the Ta displacements were given in the paper, it's not entirely clear to us where the information by the referee comes from and how was their quantitative appraisal carried out by the referee, but we hope that the clarification of the figure and Ta displacements above resolve the issue. The positional disorder of S atoms was not disregarded, since S atoms are included in the fits, but they are omitted from the images for clarity.

The very significant contribution of Ta-S pair correlations (together with the non-negligible contribution of S-S correlations !) to fine features of the experimental PDF data, which could be affected by S disorder considerably, is not accounted for either.

Author's response: The Ta-S pair correlations and S-S correlations were actually fully taken into account by the model fits. This is now clearly stated in the revised documents.

The overall impression is that the authors tend to employ overly simplistic atomic configurations to characterize the complex structure of 1T-TaS₂ and evoke a phase segregation scenario to explain the resulting large discrepancies between the model computed and experimental PDF data. This leaves a lot of room for misinterpretation of the PDF data. The authors need to explore more realistic model atomic configurations that explain the PDF data well and then try to explain the remaining small discrepancy between the model and computed PDF data, if any, and not the other way around.

Author's response: The above comments seem to be a result of multiple misunderstandings, possibly resulting from incomplete model description, as explained in the previous responses above. We do not use an oversimplistic single-layer model, as assumed by the referee, but a multilayer model, involving a S-Ta-S slab and atoms in adjacent slabs, as much as is justified by the method, with inclusion of S and Ta, and completely realistic displacements for Ta. Since we focus on the Ta-subsystem, we mainly discuss Ta layers where the largest contributions to the PDF signal are. We note that the model does account for other pair correlations as well, particularly Ta-S, as per comments above. Starting from any Ta atom, there are a number of Ta atoms within 1 nm which are NOT in-plane. How exactly this is handled by the fits is explained in the revised description. The physical picture which emerges from these considerations are presented in Figs. 3 and 4. The model however is more complex. A detailed analysis of the possible misinterpretation through over-parametrization is included in the supplement of the revised manuscript: various model options are examined, justifying quantitatively our choice of model. As already stated, our fits are a reasonable compromise between over-parametrization (model too complex for the data range) and underfitting (model inadequacy). The result is a trustworthy determination of the local polaron structure, which is the main aim of the present paper and its conclusions. Using too many parameters without clear physical justification results in a cloudy physical picture, which only leads to undesirable confusion of the fundamental issues.

The 'phase segregation' is actually a discommensuration (which is a very different physical concept) for which there is a vast amount of supporting literature, and its use is fully justified. The two phases used represent two distinct local environments present in the NC phase. In the IC phase, the modulation is incommensurate, which is approximated as a mixture of two phases. A fully incommensurate phase has an infinite unit cell, which is intractable even if approximated as a quasi-periodic structure.

Furthermore, the authors tend to attribute evolution of selected PDF features (Fig. 4a,b, Suppl. Fig 3d and Suppl. Fig. 5b,c) to evolution of assumed structural features of 1T-TaS₂ without presenting evidence that their assumptions are correct.

Author's response: The referee is correct for bringing up this aspect of our analysis, and we are grateful for bringing this deficiency to our attention. The inter-layer PDF peaks that we considered are assigned based on undistorted P-3m model, as reported in literature for 300 K. In this, interlayer spacings along the stacking direction are identified as multiples of c-axis lattice parameter, including ~5.9 Å which clearly corresponds to intra-bilayer separation. The ~13.1 Å feature in undistorted P-3m structure corresponds to a well-defined off-axis Ta-Ta contribution connecting two Ta atoms which reside in layers that are 2 interlayer spacings apart (next nearest neighbor interlayers). This interatomic distance in the undistorted structure, verified by a distance search-match, is dominated by Ta-Ta contributions, as shown in Fig 1e. To clarify this and aid plausibility of interpretation, we added text in respective figure captions, and provided further detail in Methods section. Although there is currently no model that could independently corroborate this assignment, which is required for an unambiguous assignment, we elaborate on the reasoning behind our interpretation in Methods, observation of which will hopefully prompt further studies. We softened the language to reflect the provisional character of our interpretation.

The authors also make conclusions based on miniscule changes in the PDF data (e.g. Suppl. Fig. 5b,c) that are comparable in magnitude to Fourier transformation ripples and data noise (compare with the low-r part of the PDF data in Fig. 2c,f and Suppl. Fig. 5a). The authors should convince the readers that such changes are real structural features of 1T-TaS₂ and not an experimental artifact.

Author's response: We thank the referee for reminding us of this important concern, which we have neglected in the original submission. However, we do note that the suggestion that comparable magnitude of the differential signal shown in Suppl. Fig. 5 and in Fig. 2c,f implies possible experimental artifact, albeit of a general concern in PDF analysis, is somewhat misleading in the context of the analysis that we have done here. We emphasize that the character of these differentials is very different. In Fig. 2c,d the differential corresponds to data-minus-model, whereas in Fig. 5 the differential is between two datasets. First, we note that in our measurements the data collection time was carefully evaluated by carrying out signal/noise tests. In these, satisfactory statistics was achieved already after 10 seconds of collection. The data used in the analysis were collected for 120 seconds. For samples scattering well, such as TaS₂, the setup at 28-ID-1 beamline at NSLS II is known for giving exceptionally noise-clean data. Furthermore, data are corrected for electronic noise in the detector, which is part of standard procedure at the reduction stage. Second, the main contribution to the differentials (low-r region) obtained for model fits comes from the fact that inaccuracies in the model impact accuracy in calculation of Fourier Transform ripples (primarily their position), leading to the observed differentials. Third, any noise propagation would affect a range of interatomic distances and would not appear under specific subgroup of PDF peaks. Fourth, Fourier Transform ripples in our data are fully reproducible, and their contribution is effectively cancelled by subtraction when two data sets are compared, as can be readily verified by inspecting the low r-region preceding the nearest neighbor peak. The same is expected from any experimental noise contribution, which is for our measurements insignificant, as per above comment. Fifth, the differences in Suppl. Fig. 5 are r-dependent, occurring under some PDF peaks, but not occurring under others. The character of these

peaks is emphasized by the vertical ticks plotted underneath for Ta-Ta contributions in Suppl. Fig. 5a. We have added statements to this effect in the Suppl. Fig. 5 caption to address this concern.

In conclusion, unless my criticism is addressed properly, I may not recommend the paper for publication.

Author's response: We sincerely apologize for the incomplete description of the modeling that has led to concerns of the referee. We have addressed the referee's comments by substantially expanding the Methods section and including a detailed analysis of different approaches that justifies the use of the presented modeling in three new figures in the supplement (S6-S8). We hope that this removes the possibility of further misunderstandings and presents a convincing appraisal for the choice of model and interpretation of the fitting results. We have also emphasized more clearly the S position modeling. We hope that the answers and revisions meet with the referee's approval regarding this manuscript.

REVIEWERS' COMMENTS

Reviewer #2 (Remarks to the Author):

The revised paper appears improved in many aspects, but, nevertheless, its major conclusion concerning crystallization of polarons in 1T-TaS₂ remains largely unsubstantiated. Therefore, I may not recommend it for publication.

The major problem is in the data interpretation, which has been carried out using an overly simplistic mono (S-Ta-S)-layer model aided by mechanistically subtracting data sets from one another and then attempting to explain the not surprisingly large residual differences. The so-called “polaron signature” appearing as a PDF peak shoulder at about 3.5 Å is very likely due to S-S correlations (as evaluated by the reviewer) that constitute about 10 % of the PDF intensity but are systematically neglected by the authors. Atomic PDFs are notoriously difficult to be normalized on an absolute scale and prone to “absorb” systematic data processing errors and experimental noise. Before going into interpreting vanishingly small PDF features the authors should have convinced themselves and the readers that the features are physically sensible and not artifacts.

Overall, other than the unsupported polaron & Wigner-crystal-like state idea, the paper does not say much new about the charge density wave phases of 1T-TaS₂. Some results shown in the paper seem borrowed from literature sources without giving a proper reference, which is regrettable.

Reviewer #3 (Remarks to the Author):

MANUSCRIPT NUMBER: NCOMMS-23-08389A (415921)

JOURNAL: Nature Communications

TITLE: Crystallization of polarons through charge and spin ordering transitions in 1T-TaS₂

AUTHORS: E.S. Bozin, M. Abeykoon, S. Conradson, G. Baldinozzi, P. Sutar, and D. Mihailovic

The first review of the manuscript (the revised version)

General overview of the paper

In the manuscript entitled “Crystallization of polarons through charge and spin ordering transitions in 1T-TaS₂” by E.S. Bozin, M. Abeykoon, S. Conradson, G. Baldinozzi, P. Sutar, and D. Mihailovic, the authors study the temperature evolution of the local atomic structure of the 1T-TaS₂ compound (the member of the transition metal dichalcogenide family). They use pair distribution function technique. The main conclusion of the manuscript is that low-temperature charge order is rather driven by polaron crystallization than Fermi surface nesting nor conventional electron-phonon coupling. The presented data are impressive and its analysis provides deep understanding of the physics of investigated orderings.

The submitted paper is 17 pages long including 5 pages of 70 references, 4 figures (equivalent to 4 pages), 2 tables and a page of the front matter with title and abstract. It gives more or less about 7 pages of true text in the current formatting, what makes the length of manuscript rather appropriate. Diagrams (figures) are clear with very good quality (resolution). The abstract contains the essential information about the article and summarizes the results presented in the manuscript. The article is written in very good English. There is also attached relatively detailed supplementary information.

After careful reading of the manuscript and the authors' response to referee #2, I can recommend the manuscript for the publication in Nature Communication journal in the present form.

In my opinion the authors addressed all concerns and sufficiently supported their interpretation of the data. The points mentioned below should be addressed before the final publication of the manuscript (they are of minor importance).

Specific comments to the authors:

1) PDF abbreviation is not defined in the main text of the manuscript, where it appears for the first time (it is defined only in the “Method” section).

2) For me, as a reader, it is not clear how the phases mentioned are defined like “C phase” (sometimes written as C-phase), IC. What are “CCDW”, “NCCDW” and “ICCDW” phases from Fig. 1 and its correspondence with C and IC phases? Only NCCDW is mentioned in the text. Please clarify.

3) The authors mentioned several times “Wigner crystallization” and provides some interesting references. I would like to put their attention that such phenomenon can also occur away half-filling in strongly correlated systems [cf., e.g., Phys. Rev. B 95, 125112 (2017) and references therein].

In my opinion, the manuscript should be published in Nature Communications journal. I think that the manuscript will be very interesting for the scientist working on still very popular and important field of two-dimensional systems.

REVIEWERS' COMMENTS

Reviewer #2 (Remarks to the Author):

The revised paper appears improved in many aspects, but, nevertheless, its major conclusion concerning crystallization of polarons in 1T-TaS₂ remains largely unsubstantiated. Therefore, I may not recommend it for publication.

Author's response: We regret that the referee is unwilling to accept our responses (which the other two referees find acceptable and appropriate.)

The major problem is in the data interpretation, which has been carried out using an overly simplistic mono (S-Ta-S)-layer model aided by mechanistically subtracting data sets from one another and then attempting to explain the not surprisingly large residual differences. The so-called "polaron signature" appearing as a PDF peak shoulder at about 3.5 Å is very likely due to S-S correlations (as evaluated by the reviewer) that constitute about 10 % of the PDF intensity but are systematically neglected by the authors. Atomic PDFs are notoriously difficult to be normalized on an absolute scale and prone to "absorb" systematic data processing errors and experimental noise. Before going into interpreting vanishingly small PDF features the authors should have convinced themselves and the readers that the features are physically sensible and not artifacts.

Author's response: Unfortunately, the present report is inconsistent, which makes it difficult to respond to: In the beginning, the report states that "large residual differences" are observed in the PDF data between different temperatures, while later there is a statement saying that the observed PDF features are vanishingly small. Both cannot be true.

There is another set of confusing statements in the report regarding S-S correlations. In our response and in the paper, we state that S ions are fully taken into account by the in-plane Ta-S model. We also noted in our previous response that we also take into account the nearest inter-layer distances in the modelling. We also noted in the MS that the S signal is much weaker than the Ta. So, it is not clear which S-S correlations are not taken into account.

To make sure that the issue of S correlations is not ignored entirely, we note in the revised manuscript that "S-S correlations and steric effects may play a subservient role in the ordering, but considering that the electronic S bands are far from the Fermi level, cannot be a driving force for intra-layer ordering. Future studies may be able to reveal more details in the S ordering"

Overall, other than the unsupported polaron & Wigner-crystal-like state idea, the paper does not say much new about the charge density wave phases of 1T-TaS₂.

Author's response: We respectfully disagree. The main finding is the existence of a clear polaron signature in the high-temperature metallic phase, and the 50% drop of the PDF amplitude when the polytype transformation takes place, consistent with each other layer becoming 1H polytype. This observation is the major breakthrough from which the entire picture follows. The reviewer does not appear to have acknowledged this.

Some results shown in the paper seem borrowed from literature sources without giving a proper reference, which is regrettable.

Author's response: Unfortunately, we are not able to discern which literature sources the referee refers to. While we have made a special effort to cite all the pertinent literature in preparing this manuscript, we regret if we have inadvertently omitted something. We categorically state that we have not used any results from the literature without citing it.

Reviewer #3 (Remarks to the Author):

MANUSCRIPT NUMBER: NCOMMS-23-08389A (415921)

JOURNAL: Nature Communications

TITLE: Crystallization of polarons through charge and spin ordering transitions in 1T-TaS₂

AUTHORS: E.S. Bozin, M. Abeykoon, S. Conradson, G. Baldinozzi, P. Sutar, and D. Mihailovic

The first review of the manuscript (the revised version)

General overview of the paper

In the manuscript entitled "Crystallization of polarons through charge and spin ordering transitions in 1T-TaS₂" by E.S. Bozin, M. Abeykoon, S. Conradson, G. Baldinozzi, P. Sutar, and D. Mihailovic, the authors study the

temperature evolution of the local atomic structure of the 1T-TaS₂ compound (the member of the transition metal dichalcogenide family). They use pair distribution function technique. The main conclusion of the manuscript is that low-temperature charge order is rather driven by polaron crystallization than Fermi surface nesting nor conventional electron-phonon coupling. The presented data are impressive and its analysis provides deep understanding of the physics of investigated orderings.

The submitted paper is 17 pages long including 5 pages of 70 references, 4 figures (equivalent to 4 pages), 2 tables and a page of the front matter with title and abstract. It gives more or less about 7 pages of true text in the current formatting, what makes the length of manuscript rather appropriate. Diagrams (figures) are clear with very good quality (resolution). The abstract contains the essential information about the article and summarizes the results presented in the manuscript. The article is written in very good English. There is also attached relatively detailed supplementary information.

After careful reading of the manuscript and the authors' response to referee #2, I can recommend the manuscript for the publication in Nature Communication journal in the present form. In my opinion the authors addressed all concerns and sufficiently supported their interpretation of the data.

Authors response: We are grateful to the referee for a very complimentary report, particularly for recognizing the importance of the paper. We also thank the referee for examining the issues raised by referee #2 and our responses.

The points mentioned below should be addressed before the final publication of the manuscript (they are of minor importance).

Authors response: We are grateful the careful reading of the manuscript and the specific comments

Specific comments to the authors:

1) PDF abbreviation is not defined in the main text of the manuscript, where it appears for the first time (it is defined only in the "Method" section).

Authors response: Corrected.

2) For me, as a reader, it is not clear how the phases mentioned are defined like "C phase" (sometimes written as C-phase), IC. What are "CCDW", "NCCDW" and "ICCDW" phases from Fig. 1 and its correspondence with C and IC phases? Only NCCDW is mentioned in the text. Please clarify.

Authors response: Our apologies. The notation was defined and unified throughout the document.

3) The authors mentioned several times "Wigner crystallization" and provides some interesting references. I would like to put their attention that such phenomenon can also occur away half-filling in strongly correlated systems [cf., e.g., Phys. Rev. B 95, 125112 (2017) and references therein].

Authors response: We were not aware of this reference, but it is very pertinent in showing that the polaron crystallization effect may be more general and may apply to other systems as well. We included a sentence to this effect in the conclusions, with the additional reference mentioned above.

In my opinion, the manuscript should be published in Nature Communications journal. I think that the manuscript will be very interesting for the scientist working on still very popular and important field of two-dimensional systems.